# Smart Chip Technology for the Control and Management of Invasive Plant Species: A Review

**DOI:** 10.3390/plants14101510

**Published:** 2025-05-18

**Authors:** Qaiser Javed, Mohammed Bouhadi, Smiljana Goreta Ban, Dean Ban, David Heath, Babar Iqbal, Jianfan Sun, Marko Černe

**Affiliations:** 1Institute of Agriculture and Tourism, Karla Huguesa 8, 52440 Poreč, Croatia; qaiser@iptpo.hr (Q.J.); mohammed@iptpo.hr (M.B.); smilja@iptpo.hr (S.G.B.); dean@iptpo.hr (D.B.); 2Jožef Stefan Institute, Jamova Cesta 39, 1000 Ljubljana, Slovenia; david.heath@ijs.si; 3School of the Environment and Safety Engineering, Jiangsu University, Zhenjiang 212013, China; babar@ujs.edu.cn (B.I.); zxsjf@ujs.edu.cn (J.S.)

**Keywords:** artificial intelligence, biosensors, internet of things, invasive plant control, precision agriculture, remote sensing, sustainable ecosystem management

## Abstract

Invasive plant species threaten biodiversity, disrupt ecosystems, and are costly to manage. Standard control methods, such as mechanical and chemical (herbicides), are usually ineffective and time-consuming and negatively affect the environment, especially in the latter case. This review explores the potential of smart chip technology (SCT) as a sustainable, precision approach tool for invasive species management. Integrating microchip sensors with artificial intelligence (AI) into the Internet of Things (IoT) and remote sensing technology allows for real-time monitoring, predictive modelling, and focused action, significantly improving management effectiveness. As one of many examples discussed herein, AI-driven decision-making systems can process real-time data from IoT-enabled environmental sensors to optimize invasive species detection. Smart chip technology also offers real-time monitoring of invasive species’ life processes, spread, and environmental effects, enabling artificial intelligence-powered eco-friendly control strategies that minimize herbicide usage and lessen collateral ecosystem damage. Despite the potential of SCT, challenges remain, including cost, biodegradability, and regulatory constraints. However, recent advances in biodegradable electronics and AI-driven automation offer promising solutions to many identified obstacles. Future research should focus on scalable deployment, improved predictive analytics, and interdisciplinary collaboration to drive innovation. Using SCT can help make invasive species control more sustainable while supporting biodiversity and strengthening agricultural systems.

## 1. Introduction

The rapid spread of invasive plant species is a major ecological and economic challenge worldwide, as they alter soil chemistry and disrupt native biodiversity by outcompeting indigenous plants [1,2]. Invasive species are defined by the FAO (2015) as being “Species that are non-native to a particular ecosystem and whose introduction and spread cause, or are likely to cause, socio-cultural, economic, or environmental harm, or harm to human health” [3], while the Convention on Biological Diversity (CBD) describe invasive alien species as “species whose introduction and/or spread outside their natural past or present distribution threatens biological diversity” [4] and the International Union for Conservation of Nature (IUCN) as “Animals, plants or other organisms introduced by man into places out of their natural range of distribution, where they become established and disperse, generating a negative impact on the local ecosystem and species” [5]. What is known is that invasive species exploit environmental changes, climate variability, and human-mediated disturbances to establish ecosystem dominance [6,7,8]. Conventional management strategies, including mechanical removal, chemical control (herbicides), and biological control, often fall short due to the cost of implementation, labour-intensive processes, and unintended environmental consequences such as soil degradation, water contamination, and harm to non-target species [9].

Heringer et al. [10] estimated that the cumulative economic cost of invasive species in urban areas reached approximately US$326.7 billion between 1965 and 2021, while other studies estimate that global losses due to invasive species across all sectors likely exceed US$2.2 trillion [11,12,13,14]. Two good examples are Japanese knotweed (*Fallopia japonica*, syn. *Reynoutria japonica*), which has caused significant infrastructure damage in the UK, leading to estimated losses of £166 million per year [15], and common ragweed (*Ambrosia artemisiifolia* L.), which damages agriculture and human health. Ambrosia-related agricultural losses are estimated at €13.5 million per year, while Ambrosia-induced allergies cost €7.4 billion annually in Europe [16].

Addressing the problem of invasive species and their impact on agriculture, forestry, and nature reserves will require innovative, technology-driven solutions for better detection, monitoring, and targeted eradication efforts [17]. Integrating advanced digital technologies such as embedded chip systems, biosensors, and AI offers a ground-breaking approach to invasive species management to achieve sustainable control.

Smart chip technology (SCT) has already demonstrated its potential in precision agriculture, forestry, and conservation [18]. For example, SCT can be used to monitor, visualize and generate digital data to control the application of resources and improve productivity. They are also used in precision farming, such as helping to determine precisely when and how much water a given crop needs or in devices that monitor nutrient levels and disease outbreaks [19,20]. In Europe, the Tyndall National Institute is conducting a collaborative research project called ‘HALY.ID’ with Teagasc. This project aims to monitor the highly invasive insect species *Halyomorpha halys* (Stål) (brown marmorated stink bug). Here, AI and drones automate the monitoring process, replacing traditional manual monitoring methods with AI technology [21]. Importantly, when combined with AI-driven predictive modeling, these chips support targeted interventions, helping land managers optimize decision-making [22]. A similar strategy could be adapted for managing invasive plant species, where bioengineered microchips embedded in targeted plants would track their physiological traits and spread dynamics, offering valuable insights for controlling their environmental impact.

Integrating SCT into ecological conservation, sustainable agriculture, and biodiversity restoration offers transformative potential. This review focuses on the role of SCT in invasive species management, highlighting its advantages, challenges, and future directions. Smart sensors installed in invasive species such as *Eichhornia crassipes* (Mart.) Solms., (water hyacinth) have been used to gather real-time information on growth rate and nutrient uptake [23]. This information can help land managers implement timely eradication strategies before real environmental harm occurs. In another study, sensors connected to an invasive plant were used to detect its tolerance to biotic and abiotic stresses, helping predict its survival under climate change scenarios [24]. Drones equipped with remote sensors could target affected regions with eco-friendly herbicide sprays or biological control agents, reducing the reliance on large-scale chemical treatments [25]. Invasive species eradication programs in Australia and the United States, for instance, are increasingly exploring automated drone-based monitoring systems, which could be further enhanced by integrating smart chips for data collection, allowing real-time management decisions [26,27].

Despite SCT’s potential, many obstacles remain, such as the cost of deployment, especially on a large scale, issues relating to the environmental risks of electronic waste if disposal is not correctly managed, and regulatory challenges regarding introducing such devices in the natural environment. However, ongoing research in nanotechnology and biodegradable electronic materials could result in eco-friendly smart chips. Researchers are developing bioresorbable electronics based on micro-chip technology capable of transmitting plant health data before naturally decomposing into the environment [28,29,30].

## 2. Smart Chip Technology: An Innovative Approach

A smart chip is an integrated circuit (IC) that can process data, interact with its environment, and communicate with other devices. It is designed with embedded sensors, processors, and communication technology, enabling it to collect, process and transmit data for various applications and make decisions or take actions based on their input, often in real-time [31]. Smart chip technology is a broader term for using these chips in various applications, including environmental monitoring, agriculture, healthcare, and more (Figure 1). In invasive species management, smart chip technology can include sensors to detect invasive species or environmental conditions and communicate data to a central system for analysis and decision-making. Integrating SCT into invasive plant management represents a novel approach to crop management. Farmers can use embedded SCT to monitor their real-time physiological responses, growth dynamics, and environmental impact (Figure 1). These chips can collect data on plant health indicators such as moisture levels, nutrient uptake, and stress responses, allowing for early detection and targeted intervention [32,33].

Chip-based systems have already demonstrated their effectiveness in monitoring crop conditions by providing farmers with real-time data on soil moisture and nutrient availability, enabling optimized irrigation and fertilization strategies [34]. A comparable approach can be applied to invasive species, where chips embedded in invasive plants can track their spread patterns, aiding in developing precision control measures.

Bioengineered smart chips, in conjunction with biosensors, can also detect biochemical changes and trigger targeted responses, such as the release of allelopathic compounds, through embedded reservoirs or actuator technologies. Bioengineered smart chips typically operate as part of an integrated biosensor system. These chips embedded with biosensors, often based on enzymes, antibodies, or nucleic acid probes, are specifically designed to detect biochemical signals such as the release of allelopathic compounds, stress-induced metabolites, or hormonal changes in invasive plant species [35]. After detecting the targeted biochemical markers, the chip transmits signals to a processing unit via wireless or microcontroller-based systems. This ability allows for real-time monitoring and potentially triggers pre-programmed responses, such as activating countermeasures or alerts for manual intervention [35,36]. This information can be used to prioritize removing high-impact invasive species before they cause significant damage and for large-scale tracking and predictive modeling to identify high-risk areas for early interventions [37].

In smart agriculture, IoT-based sensors are widely used to detect crop diseases and environmental stressors [38,39]. For example, in vineyards, AI-integrated biosensors are being used to monitor the presence of fungal infections such as powdery mildew, alerting farmers before significant damage occurs [40]. Similarly, chip-based technology is used in citrus orchards to detect Huanglongbing (citrus greening disease), allowing for removing infected trees before the disease spreads [41]. Biosensors also have the potential to identify genetic markers unique to invasive species, enabling their precise detection even at early growth stages. Moreover, drone-assisted monitoring systems combined with smart chip-based data collection and AI autonomously target and treat affected areas using eco-friendly herbicide sprays or biological control agents. Detailed information about different types of sensors embedded in smart chips and their application is given below (Table 1), while specific commercial and research-based sensor models and chips commonly used in weed control and precision agriculture are provided in Appendix A.

## 3. Smart Chip-Enabled Invasive Plant Management

Real-time data collection is crucial for the early detection and ongoing monitoring of invasive plants (Figure 2). Technologies such as satellite imagery, drones, and ground-based sensors equipped with chips allow continuous tracking of invasive species across large areas, providing up-to-date insights into their spread and environmental impact [61,62].

For instance, agricultural landscapes, where crops like corn and soybean are frequently threatened by invasive weeds such as *Amaranthus tuberculatus* (Moq.) (waterhemp) and *Amaranthus palmeri* S. Watson [63], benefit from satellite-based monitoring systems. These systems enable farmers to identify affected areas quickly and prevent further spread in riparian zones where Japanese knotweed is a persistent invasive species. Drones equipped with GPS and sensors could monitor its spread along riverbanks, allowing land managers to develop targeted control strategies [64] similar to how Australia uses drones equipped with infrared sensors to locate and target the invasive Cane toads (*Rhinella marina Linnaeus*, 1758), delivering traps or biological agents to control the population with minimal human intervention [65].

Real-time data collection helps optimize resource allocation and response times. Recent research in Slovenia developed an automatic detection system for Japanese knotweed using a One-Class Support Vector Machine (SVM) connected to multiple Sentinel-2 satellites. The input data for this system included red-green-blue band composites, infrared bands (IR), and additional satellite images using the Normalized Difference Vegetation Index (NDVI) and Enhanced Vegetation Index (EVI). The K-means algorithm was used on the IR band to group the samples. As a result, 83% accuracy was recorded on the aerial images, and 90% was recorded on satellite data for detecting Japanese knotweed. Such information made it possible to detect Japanese knotweed at the municipality level and can help understand the distribution of this species [66]. Similarly, Catala-Roman et al. [67] used an unmanned aerial system (UASYS) equipped with neural networks (CNNs) based on YOLOv10 for detecting and geo-locating two invasive species, *Araujia sericifera Brot.* and *Cortaderia selloana* in the citrus orchard. The system significantly enhanced productivity, reduced the costs associated with manual weeding in organic farming and promoted environmental sustainability, offering a scalable method for organic and conventional agriculture.

In addition to the widespread use of satellites, drones, and AI-based detection methods, several innovative smart systems have emerged, each offering unique solutions for monitoring invasive species. For example, Nature Metrics utilizes environmental DNA (eDNA) analysis to detect invasive species in soil, water, or air samples, providing a non-invasive, highly accurate monitoring method [68]. Similarly, Trellis Data has developed AI-powered cameras designed for port biosecurity inspections, capable of identifying small invasive pests on cargo containers, thus reducing the need for time-consuming manual checks [69]. Meanwhile, Wildlife Drones offers a radio-telemetry system integrated with drones, allowing real-time tracking of up to 40 radio-tagged invasive animals across difficult terrain, which enhances targeted control efforts [70]. Fulcrum’s customizable mobile app for field data collection simplifies invasive plant species monitoring by enabling geotagged photos, offline data entry, and cloud-based synchronization for easier reporting. SGS has developed global eDNA-based services, including sample collection, DNA sequencing, and species identification, facilitating early detection of invasive species from a single environmental sample [71]. All these technologies show the growing trend toward precision, automation, and scalability in invasive species management. They also offer a range of options that vary in spatial scale, deployment costs, and application.

AI-driven analysis and predictive modelling further enhance invasive species management by enabling the processing of large datasets. Analyzing real-time and historical data, such as temperature, soil moisture, and land cover, where and when an invasive species might spread could be predicted, and areas for treatment could be prioritized. This information will allow land managers to take proactive measures before any invasion becomes uncontrollable. For example, the spread of *Pueraria montana* var. lobata (Willd.) (kudzu), an invasive plant in the south-eastern U.S., has been studied using AI models incorporating weather data and land-use patterns [72]. Similarly, AI models are being used to predict the spread of *Ailanthus altissima* (Mill.) (tree-of-heaven) in forest ecosystems based on satellite imagery, ground surveys, and climate data [73].

The application of automated removal strategies, including precision herbicide application and drones, has already dramatically improved the efficiency of invasive plant control [74,75]. For example, the application of precision herbicides using AI-enabled drones allows farmers to target invasive weeds in vegetable crops like potatoes [76,77], threatening crop health while sparing the surrounding plants. Reducing chemical usage minimizes runoff into nearby water systems [78]. The system also ensures that only affected zones are treated, significantly improving management and minimizing waste. In another study, the results showed that accurate and timely maps of *Sorghum halepense* (L.) Pers. (Johnsongrass) patches in maize can be key to achieving site-specific and sustainable herbicide applications for reducing spraying herbicides and costs [79].

Drones could play a key role in automated removal of invasive plant species, with high-resolution cameras and AI algorithms enabling rapid identification and mapping of invasive species [25]. In fields, invasive species like *Chromolaena odorata* (L.) R.M.King & H.Rob. (Siam weed) can be identified early by drones equipped with high resolution cameras [25]. Precision herbicide application can also minimize the impact on surrounding vegetation [80,81]. These systems can distinguish between target and non-target species, typically using Red, Green, and Blue (RGB) cameras, hyperspectral imaging, multispectral sensors, and machine learning models [82]. RGB cameras are considered a good method for weed identification, which uses physical characteristics such as shape, colour, and size [83]. Despite its wide application, RGB cameras are currently limited in identification, especially when the alien and indigenous species have similar leaf shapes and sizes during the early growth stage [84].

Ferreira et al. [85] grouped image pixels rendering similar colour and spatial closeness using simple-linear-iterative-clustering, with the images being divided into different sections containing multiple soy and weed leaves using this technique. Then, CNNs were used for classification, giving the results with 99.5% higher classification accuracy than conventional machine learning (ML) classifiers, i.e., support vector machines.

Hyperspectral imaging provides comprehensive details about crop vegetation by capturing a wide range of spatial and spectral information. Hyperspectral imaging uses images in narrow bands with a visible light range of 400–1000 nm and 900−1700 nm near-infrared range of the electromagnetic spectrum [86]. This information helps to identify and differentiate between weed and crop species based on the chemical information, i.e., chlorophyll content and water absorption present within the cellular structures of plants because the chemical information is inherently unique among plant species [82]. At the same time, hyperspectral imaging generates large amounts of data, which also contains redundant data that can impact the spectral data. Machine learning and deep learning (DL) techniques offer promising solutions to these challenges.

In ML and DL techniques, discrete responses are used, and every pixel in the image is classified into a distinct class for weed identification. Algorithms then identify patterns, relationships, and features within the input data by iteratively adjusting predetermined parameters [86]. Mensah et al. [82] achieved accuracy between 75% and 95% by using ML and DL over traditional classifiers such as Partial Least Square Discriminant Analysis, Random Forest, Spectral Angle Mapping, Maximum Likelihood Classifier, Support Vector Machine, and Bayesian classifier have been utilized for weed identification. However, the same approach can be employed with the latest developments in hyperspectral remote sensing to differentiate between crops and invasive plant species in farmlands.

Robotic systems, in combination with SCT, are also being investigated for invasive species management [87]. These robots can, through the use of machine learning, autonomously identify and remove invasive plants with a strimmer of a robotic hand. This ability is especially beneficial in labour-intensive or hazardous environments [88]. Bakker et al. [89] used autonomous robots with real-time kinematics global positioning systems to complete the inter-row hoeing in corn crops without damage to the surrounding plants. Nørremark et al. [90] recorded 91% success rates by removing numerous inter-row weeds using a cycloid robotic, while Kunz et al. [91] used Robocrop, a camera-guided inter-row hoeing device and noted an 89% reduction in weed density in soybean crops and 87% in sugar beet crops. Other devices, such as AgBotII, are capable of accurately identifying up to 90% of selected weeds, including wild oats (*Avena fatua* L.) and common sowthistle (*Sonchus oleraceus* L.) [92]. BoniRob, produced by Bosch Deepfield Robotics, showed 94% success in monitoring various weeds [93]. These robots help reduce the need for costly and potentially harmful chemical herbicides (e.g., Glyphosate) while improving efficiency.

## 4. Challenges and Prospects of Smart Chip-Enabled Invasive Plant Management

Chapman and Brooke (2021) investigated various methods, including hand pulling, herbicide application, mowing, cutting, and daubing (Figure 3), to eradicate invasive plant species at the Santa Lucia Preserve [94]. They successfully removed invasive plants without impacting native species and disrupting the habitat. While their techniques succeeded in controlling the invasive species, the more advanced technologies, drones, and AI-driven systems show great promise.

Despite the numerous benefits, chip-enabled invasive plant management faces challenges, including cost, environmental impact, and ethical considerations. Among these, the main barrier remains the high cost of its implementation and the initial investment required for drones, sensors, AI-driven systems, and robotics, particularly for small-scale farmers, especially those without access to sufficient capital or subsidies. Also, they acquire technical knowledge in fly drones and robotics, software and data processing, maintenance and robustness of the equipment. In numerous studies, UASYS has been used to acquire image data for successfully sensing and monitoring invasive plants. However, these studies used high-end UASYS (e.g., Michez et al. [95]) and commercial image processing software, which cost 3000 US dollars or more for a single license. Fixed-wing UASYS, as an alternative, is being used widely with multirotor drones. SCompaniessuch as senseFly, QuestUAV, Gatewing, and Trimble offer ready-to-fly fixed-wing UASYS with full autopilot control and camera sensors. The system costs less than 2000 US dollars. Although this approach does not charge extra labour costs, it can be operated easily with only basic knowledge of image processing and GIS [96].

In another study, Takekawa et al. [97] used UASYS to monitor pepper weed more effectively (*Lepidium latifolium* L.) in Suisun Marsh in northern California, USA. In Suisun Marsh, a team of hand backpack sprayers commercially costs USD 456.95/ha to spray the herbicides, whereas tractor and spraying helicopter-spraying costs 97.50/ha USD and 140.45/ha USD. However, Takekawa et al. [97] reported operational costs of USD 36.79 /ha for the UASYS system, mainly attributed to technician labor, compared to the higher costs associated with drone spraying, which includes chemical mixing, drone operation, and logistical management. The company “Leading Edge Aerial Technologies, Inc.” commercially charged USD 125–150/ha (estimated) for spraying with drone application, including equipment and pilot, but it is still cheaper than contracting with backpack sprayers.

The ethical considerations surrounding autonomous technologies in invasive species management involve issues such as access to public and private lands and the potential impacts on agricultural land abandonment. In many regions, land abandonment occurs due to unfavourable cultivation conditions, low farm profitability, and broader socio-economic challenges [98]. These abandoned areas often become a good hotspot for invasive plant species, aggravating biodiversity loss, increasing fire risk, and accelerating soil erosion [99]. Although using AI-driven drones combined with remote sensing and GIS offers a promising solution for spotting, identifying, and managing invasive plants in such areas, implementing such SCT raises concerns about land rights, privacy, and regulatory oversight [99]. In community farms or gardens, questions may arise about the ethical implications of automation in decision-making, such as which species should be removed and the surety not to remove beneficial plants. For this reason, the ethical and practical deployment of these technologies should align with broader conservation goals while respecting landownership rights and local ecological dynamics.

Expanding these technologies to invasive species management presents another exciting opportunity. Many of the tools for controlling weeds in natural environments can be adapted to manage invasive plant species. For instance, AI and sensor networks in wheat fields can work with robotic systems to detect and uproot invasive species like wild oats competing with wheat for nutrients [100]. In rice paddies, the precision herbicide application systems developed for controlling waterhemp [75] could be applied to manage *Lemna* L. (duckweed), which can overtake rice fields if left unchecked [101]. In addition, existing models powered by AI used for invasion prediction for crops infested by Cheatgrass (*Bromus tectorum* L.) [102] may be modified for other invasive plants, assisting farmers in knowing when action is needed.

AI-based technologies could also greatly benefit ecological restoration [103]. Automating invasive species removal makes large-scale restoration projects more feasible [104]. Robotics and drones aiding in the restoration of native flora, such as in the U.S., the efforts to restore native grasslands, automated systems could target and remove invasive *Lespedeza cuneata* (Dum.Cours.) G.Don, allowing native grasses to regrow [105]. In wetland (invaded by *Lythrum salicaria* L. (purple loosestrife, [106]) restoration, these systems could be used to remove invasive purple loosestrife, facilitating the restoration of critical wetland habitats for species like waterfowl and native fish. The same technology could tackle bindweed in grapevine fields, preventing it from spreading, smothering the vines and reducing competition for nutrients and water [107]. Integrating SCT with remote sensing, robotics, and AI-driven predictive models holds significant potential for the future.

## 5. Conclusions

### 5.1. Summary of Key Findings

This paper looked at the ability of SCT to serve as a new methodology for invasive plant species management. Embedded microchip sensors with AI, IoT, and remote sensing technologies can enhance real-time monitoring and precision control measures. For example, they will allow land managers to monitor the invasive species’ physiological traits, spread dynamics, and environmental interactions, making remedial measures more targeted and effective. Such proactive measures will allow early detection combined with rapid response, thereby reducing the negative impacts of broad-spectrum herbicides. The benefits of smart chip-enabled monitoring systems are already being used in precision agriculture, forest management, and ecological monitoring. Case studies show that using GPS sensors integrated with bioengineered microchips and drones if applied to invasive species management, could significantly improve the efficiency and sustainability of invasive species management. However, challenges include unfavourable implementation costs, microchip biodegradability, environmental issues (e.g., chemical leaching, E-waste accumulation and wildlife interference) and controlling policies.

### 5.2. Call for Further Research and Technology Development

While some technologies have tried to incorporate smart chips in managing invasive species with little success, optimizing the technology’s integration, cost, and environmental impact requires further investigation. This task should include looking into more affordable production methods, as well as other strategies that can be broadly implemented so that small farmers and conservation groups can benefit. As a starting point, future research should focus on developing more sophisticated AI models capable of predictive mapping so that invasions can be anticipated and addressed before they have started. Automating the drone-assisted precision treatment processes and data gathering will further improve the control of invasive species by efficient detection and mapping, targeted treatment application and high-resolution data collection and analysis. Achieving these goals will require a multidisciplinary approach by ecologists, engineers, and policymakers to design regulatory options and best practices to mitigate the risks associated with these new technologies.

## Figures and Tables

**Figure 1 plants-14-01510-f001:**
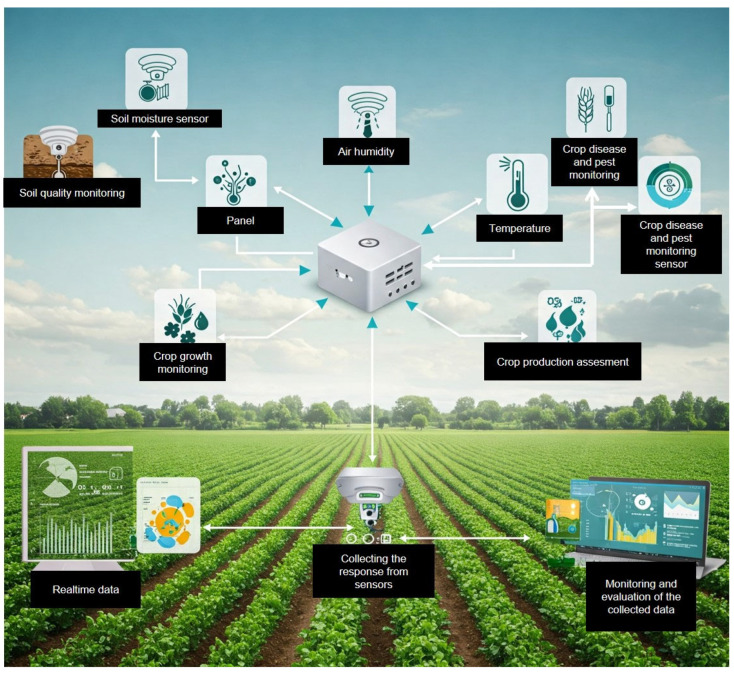
Conceptual representation of a smart farming system integrating IoT devices and various sensor technologies to monitor key environmental and crop health factors for improved agricultural outcomes. The system includes soil moisture and quality sensors, air humidity and temperature sensors, and crop disease and pest monitoring sensors. These sensors help monitor crop growth, assess production, and detect early signs of stress or infestation. Data collected from the field is transmitted in real-time to central processing units and digital analytics platforms, enabling informed decision-making and sustainable precision agriculture practices.

**Figure 2 plants-14-01510-f002:**
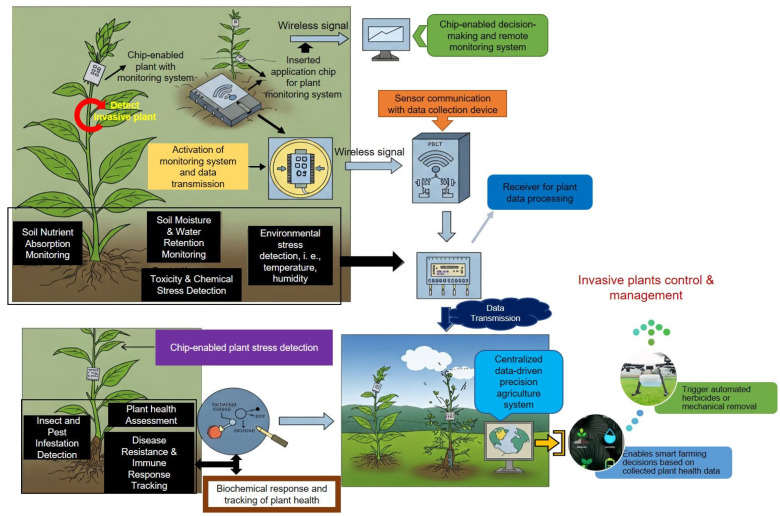
Smart plant monitoring system using chip-enabled technology for invasive species management. The diagram illustrates how embedded sensors monitor plant health, detect invasive species, and assess environmental stressors. The system enables real-time data collection and remote decision-making by tracking soil conditions, pest infestations, and chemical responses. This technology supports early detection and precise management of invasive species.

**Figure 3 plants-14-01510-f003:**
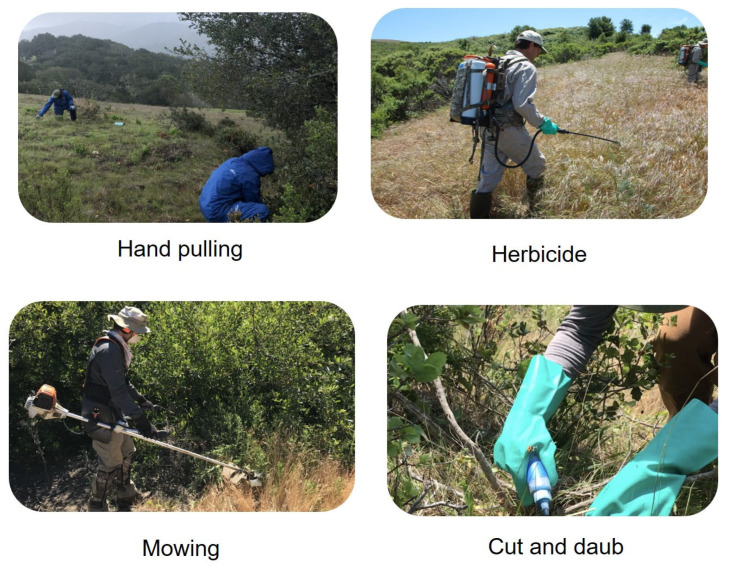
The ways to treat invasive species in natural areas at Santa Lucia Preserve, the Conservancy practices minimally intrusive methods for maximum effect. They use four tried and true methods to target invasive plants: hand pulling, spot-spraying herbicide, spot-mowing, and cut and daub [94].

**Table 1 plants-14-01510-t001:** Types of sensors, applications, and data collection methods in smart chip systems for precision agriculture and environmental monitoring.

Sensor Type	Application in Field	Data Collected	Data Collection Method	Studied Crop/Plant/Species	References
Biosensors	Detect plant diseases, stress responses, and allelopathic interactions.	Enzyme activity, secondary metabolites, stress markers	Embedded microchips with biochemical detection, wireless transmission	*Zea mays* L. (Maize), *Oryza sativa* L. (Rice)	[42]
GPS Sensors	Track the movement and spread of invasive plant species.	Geolocation, plant movement patterns	GPS-embedded chips, satellite-based tracking.	*Parthenium hysterophorus* L., *Lantana camara* L	[43,44]
IoT-Based Smart Sensors	Monitor soil parameters for precision agriculture.	Soil moisture, temperature, pH, salinity, humidity	Wireless IoT network, real-time monitoring.	*Triticum aestivum* L. (Wheat), *Gossypium hirsutum* L. (Cotton)	[45,46]
Electrochemical Sensors	Measure soil nutrient levels and heavy metal contamination.	pH, nitrogen, phosphorus, potassium levels, contaminants.	Integrated electrochemical probes, automated soil analysis	*Brassica napus* L. (Canola), *Glycine max* (L.) Merr. (Soybean)	[47,48]
AI-Integrated Image Sensors	Identify plant diseases and classify species using machine learning.	Leaf shape, disease symptoms, chlorophyll content	AI-driven image analysis, UAV and drone-based monitoring	*Solanum lycopersicum* L. (Tomato), *Vitis vinifera* L. (Grapevine)	[49,50]
Dendrometers	Monitor tree and shrub growth rates.	Stem diameter, biomass accumulation	Sensor-equipped microchips attached to plant stems.	*Quercus robur* L. (Oak), *Pinus sylvestris* L. (Scots Pine)	[51]
Thermal Infrared Sensors	Detect plant stress and drought conditions.	Leaf temperature, transpiration rates, canopy stress	UAV-based infrared thermography, real-time heat mapping	*Citrus × sinensis* (L.) Osbeck. (Orange), *Capsicum annuum* L. (Pepper)	[52,53]
Hyperspectral Sensors	Differentiate between native and invasive species based on spectral signatures.	Reflectance indices, vegetation health data	UAV and satellite-based spectral imaging.	*Alternanthera philoxeroides* (Mart.) Griseb. (Alligator weed)	[54,55]
Microfluidic Sensors	Analyze plant sap flow, water use efficiency	Xylem conductivity, nutrient transport rates	Embedded chip with microfluidic flow analysis	Woody plants	[56]
Chlorophyll Fluorescence Sensors	Detect photosynthetic efficiency and plant health	Fv/Fm ratio, electron transport rate	Portable fluorescence imaging, automated leaf-level analysis	*Phaseolus vulgaris* L. (Common bean), *Coffea arabica* L. (Coffee)	[57]
Drones with Multispectral Cameras	Conduct large-scale monitoring of crop health and invasive weeds	NDVI, LAI, canopy cover, disease identification	UAV-based remote sensing, automated GIS mapping	*Cicer arietinum* L. (Chickpea)	[58]
LIDAR Sensors	Assess plant canopy structure, biomass estimation	Canopy height, vegetation density	Airborne LIDAR scanning, 3D mapping	*Populus tremuloides* Michx., *Eucalyptus globulus* Labill	[59]
Nano-Sensors	Detect chemical stressors, pollutant levels in soil and plants	Heavy metals, pesticide residues, soil contamination	Nanoscale biosensor technology, colorimetric detection	*Zea mays* L. (Maize), *Nicotiana tabacum* L. (Tobacco)	[60]
Radio Frequency Identification (RFID) Sensors	Track movements of fishes	Identification tags, plant movement patterns	RFID-embedded plant tracking, real-time monitoring	Fishes	[61]

## Data Availability

No data has been created for this manuscript.

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
