# Peer review of "Smart Chip Technology for the Control and Management of Invasive Plant Species: A Review"

_plants, 2025, doi:10.3390/plants14101510_

Round 1
Reviewer 1 Report
Comments and Suggestions for Authors
The manuscript presents a well-structured and coherent overview of the selected topic. The authors demonstrate a good command of the literature and provide a sufficiently comprehensive review of the subject. From my perspective, the manuscript does not require any substantial revisions and is suitable for publication in its current form.
Author Response
Response: We sincerely thank the reviewer for their positive and encouraging feedback. We are pleased to know that the manuscript has been well-received and that the structure, coherence, and literature coverage were considered satisfactory.
Reviewer 2 Report
Comments and Suggestions for Authors
Description of the manuscript:
This manuscript is entitled “Smart chip technology for the control and management of invasive plant species: a review”. As the self-explanatory title suggests, this manuscript is a review article. The subject nature is quite interesting because the authors attempted to link two rather different topics, namely smart chip and invasive species. The former apparent belongs to information and technology, whereas the latter certainly belongs to plant ecology. I believe that I am an eligible reviewer because I have pending publications in these two research fields.
General comments:
I have three general comments:
- First of all, smart chips or smart sensors are mentioned throughout the manuscript. Yet, I guess that the particular model of sensors or chips with extensive applications can be provided. Readers from the related research fields would be interested in taking up the sensors for further resaerch or practical uses.
- Secondly, the application of smart chips should be clarified. I believe that the authors are referring to environmentally responsive systems. In several instances in the article, the authors mentioned that different functions can be performed by chips, including the emission of alleopathetic chemicals for the suppression of invasive species. However, a chip itself cannot secrete the chemicals. Instead, it makes more sense for a system containing certain chips plus sensors to complete such task.
- Last but not least, in Table 1, different kinds of “chips” were mentioned. In fact, it may not be appropriate to call all the so-called "chips” in the table “smart chips”. The term “smart” may refer to the autonomous functionality of a system to respond and react to environmental changes. However, not every system or chip in Table 1 does that. For example, GPS sensors may only form a part of a smart system but are not smart chip nor smart sensors themselves. The authors may need clarification of concepts.
Other than my general comments, please refer to my line-by-line, specific comments for each section.
Specific comments:
Abstract
Line 12 to 15
Please shorten the first few sentences as the introduction.
Line 15 to 17
Is this the research objective? If so, please state it explicitly that this is the objective.
Line 17 to 18
IoT and AI are related but actually different concepts. The authors may have to distinguish between them, rather than habitually using them alongside each other.
Line 24
Give examples of the potential solutions.
Introduction
Line 35
Should FAO be used as the sole provider of definition? I suggest that more definitions from different sources can be engaged.
Line 49 to 50
Are the numbers applicable to the same study location, namely the UK?
Smart Chip Technology: An Innovative Approach
Line 109 to 112
The figure can be given more textual elaboration. Also, if part(s) of figure art belong to other published materials, citation should be provided.
Line 118
The operation mechanism of the bioengineered chips can be elaborated.
Smart Chip-Enabled Invasive Plant Management
Line 142
Change “like” to “such as”.
Line 145 to 172
Various examples were provided here. However, the readers may be interested in the evaluation and the comparison of effectiveness and efficiency among different smart systems. I believe that there are different so-called smart systems on the market. Still, the potential authors are looking for information about the usefulness of the systems.
Line 215 to 217
Identification accuracy is of course important. Yet, can the invasive species be effectively removed? For example, during the process of removal, escape is probable. Also, when removing invasive plant species, remaining tuber roots means that rapid return of pest species will happen.
Challenges and prospects of smart chip-enabled invasive plant management
Line 185 to 189
Other than illustrative arts, I believe that photographs of invasive species removal systems can be used. With proper citation, photographs can be inserted.
Conclusion
I have no critical comments on this section of the manuscript.
Author Response
|
Comments 1: This manuscript is entitled “Smart chip technology for the control and management of invasive plant species: a review”. As the self-explanatory title suggests, this manuscript is a review article. The subject nature is quite interesting because the authors attempted to link two rather different topics, namely smart chip and invasive species. The former apparent belongs to information and technology, whereas the latter certainly belongs to plant ecology. I believe that I am an eligible reviewer because I have pending publications in these two research fields.
|
|
Response 1: We sincerely thank the reviewer for taking the time to evaluate our manuscript titled “Smart Chip Technology for the Control and Management of Invasive Plant Species: A Review.” We appreciate your positive feedback on the interdisciplinary nature of the work and are encouraged to know that you found the attempt to bridge the fields of smart technology and plant ecology both interesting and relevant. We are grateful that you acknowledged the novelty of integrating these two distinct fields, which we believe opens new avenues for research, monitoring, and precision management in environmental sciences. Thank you once again for your kind consideration and support of this work. |
|
Comments 2: General comments: I have three general comments. |
- First of all, smart chips or smart sensors are mentioned throughout the manuscript. Yet, I guess that the particular model of sensors or chips with extensive applications can be provided. Readers from the related research fields would be interested in taking up the sensors for further resaerch or practical uses.
Response 2: Thank you for your insightful comment regarding the need for more specificity on the models of smart sensors or chips referenced in our manuscript. We have now included a comprehensive table (Table S1) in the revised manuscript (line numbers 136-138) that lists specific commercial and research-based sensor models and chips commonly used in weed control and precision agriculture. Each entry includes the application area, functionality, data collection method, and relevant source. This addition is intended to guide researchers and practitioners in selecting suitable technologies for invasive species monitoring and smart agricultural applications.
- Secondly, the application of smart chips should be clarified. I believe that the authors are referring to environmentally responsive systems. In several instances in the article, the authors mentioned that different functions can be performed by chips, including the emission of alleopathetic chemicals for the suppression of invasive species. However, a chip itself cannot secrete the chemicals. Instead, it makes more sense for a system containing certain chips plus sensors to complete such task.
Response 2: We sincerely thank the reviewer for the insightful comment and the opportunity to clarify our intended meaning regarding the application of smart chips. We agree with the reviewer’s observation that the chip alone does not possess the capability to emit allelopathic chemicals. Our intention was to refer to an integrated smart system where the chip functions as a controller or decision-making unit in conjunction with biosensors and delivery mechanisms. These systems are designed to monitor environmental cues and trigger targeted responses, such as the release of allelopathic compounds, through embedded reservoirs or actuator technologies. We have added this information in the revised manuscript (line numbers 118-120) and improved the contents.
- Last but not least, in Table 1, different kinds of “chips” were mentioned. In fact, it may not be appropriate to call all the so-called "chips” in the table “smart chips”. The term “smart” may refer to the autonomous functionality of a system to respond and react to environmental changes. However, not every system or chip in Table 1 does that. For example, GPS sensors may only form a part of a smart system but are not smart chip nor smart sensors themselves. The authors may need clarification of concepts.
Response 2: We are grateful to the reviewer for highlighting this important distinction and for pointing out the need for greater conceptual clarity regarding the use of the term “smart chips” in Table 1. We revised the caption of Table 1 as” Types of sensors, applications, and data collection methods in smart chip system for precision agriculture and environmental monitoring” to avoid the misunderstanding.
Other than my general comments, please refer to my line-by-line, specific comments for each section.
Specific comments:
Abstract
Comments 3: Line 12 to 15, Please shorten the first few sentences as the introduction.
Response 3: Thank you for the suggestion. The sentence was shortened as “Invasive plant species threaten biodiversity, disrupt ecosystems, and are costly to manage. Standard control methods, such as mechanical methods and herbicides, are usually ineffective, time-consuming, and damage the environment,” in the revised manuscript (line numbers 12-14).
Comments 4: Line 15 to 17, Is this the research objective? If so, please state it explicitly that this is the objective.
Response 4: Thank you for the suggestion. The correction has been made in the manuscript (line numbers 14-15)
Comments 5: Line 17 to 18, IoT and AI are related but actually different concepts. The authors may have to distinguish between them, rather than habitually using them alongside each other.
Line 24, Give examples of the potential solutions.
Response 5: We have revised the manuscript to clarify that while IoT (Internet of Things) involves the network of interconnected devices that collect and exchange data, AI (Artificial Intelligence) refers to the ability of systems to analyze data and make intelligent decisions or predictions based on that data. In the revised manuscript (Lines 19-21), we have also included potential solutions that arise from the integration of these technologies. For instance, AI-driven decision-making systems can process real-time data from IoT-enabled environmental sensors to optimize invasive species detection, predict outbreaks, and guide targeted control measures.
Introduction
Comments 6: Line 35, Should FAO be used as the sole provider of definition? I suggest that more definitions from different sources can be engaged.
Response 6: We thank the reviewer for this valuable suggestion. Accordingly, we have modified the contents in the revised manuscript (lines 39-45) to incorporate definitions from other key organizations, such as: Convention on Biological Diversity (CBD) described that Invasive alien species are species whose introduction and/or spread outside their natural past or present distribution threatens biological diversity.”Convention on Biological Diversity (CBD), 2002. Guiding Principles for the Prevention, Introduction and Mitigation of Impacts of Alien Species That Threaten Ecosystems, Habitats or Species. Annex to Decision VI/23. [Available online at: https://www.cbd.int/decision/cop/?id=7197] International Union for Conservation of Nature (IUCN) defined invasive alien species as “animals, plants or other organisms introduced by man into places out of their natural range of distribution, where they become established and disperse, generating a negative impact on the local ecosystem and species.”IUCN, 2000. IUCN Guidelines for the Prevention of Biodiversity Loss Caused by Alien Invasive Species. [Available online at: https://portals.iucn.org/library/node/7207]
Comments 7: Line 49 to 50, Are the numbers applicable to the same study location, namely the UK?
Response 7: We thank the reviewer for pointing out the need for clarification regarding the origin of the cited figures. To address this, we have revised the sentence to indicate that the £166 million annual loss due to Japanese Knotweed (Fallopia japonica) pertains specifically to the UK, while the €13.5 million in damages caused by Common Ragweed (Ambrosia artemisiifolia L.) refers to Europe (lines 54-57).
Smart Chip Technology: An Innovative Approach
Comments 8: Line 109 to 112, The figure can be given more textual elaboration. Also, if part(s) of figure art belong to other published materials, citation should be provided.
Response 8: Thank you for the suggestion. We have elaborated on the textual details (lines 116-122) in the figure caption in the revised manuscript. The Figure is created by the authors.
Comments 9: Line 118, The operation mechanism of the bioengineered chips can be elaborated.
Response 9: We thank the reviewer for this insightful comment. We agree that further elaboration on the operational mechanism of bioengineered smart chips will enhance the reader's understanding. In the revised manuscript (lines 130-138), we have expanded this section to clarify that bioengineered smart chips typically operate as part of an integrated biosensing system. These chips are embedded with biosensors—often based on enzymes, antibodies, or nucleic acid probes—that are specifically designed to detect biochemical signals such as allelopathic compound secretion, stress-induced metabolites, or hormonal changes in invasive plant species. Upon detecting target biochemical markers, the chip transmits signals to a processing unit via wireless or microcontroller-based systems. This enables real-time monitoring and potentially triggers pre-programmed responses, such as the activation of countermeasures or alerts for manual intervention. We have clarified that the chip itself does not perform biochemical functions but serves as the detection and decision-making hub within a responsive biotechnological system.
Smart Chip-Enabled Invasive Plant Management
Comments 10: Line 142, Change “like” to “such as”.
Response 10: The change has been made in the revised manuscript, Line 162.
Comments 11: Line 145 to 172, Various examples were provided here. However, the readers may be interested in the evaluation and the comparison of effectiveness and efficiency among different smart systems. I believe that there are different so-called smart systems on the market. Still, the potential authors are looking for information about the usefulness of the systems.
Response 11: Thank you for your valuable suggestion. There is already information presented in the manuscript about the effectiveness and efficiency among different smart systems (lines 177-192). For example, a recent research did in Ljubljana, Slovenia on automatic detection of Japanese knotweed by developing a One-Class support vector machine connecting with multiple Sentinel-2 satellite. The input data used for aerial images support in this vector machine was red-green-blue band composites and infrared bands (IR), and additionally normalized difference vegetation index (NDVI) and enhanced vegetation index (EVI) were used for satellite images. For grouping the samples, K-means algorithm was used on the IR band. As a result, 83% accuracy was recorded on the aerial images and 90% was recorded on satellite data for detection of Japanese knotweed. Such information made possible to detect Japanese knotweed on the municipality level and can be helpful to contribute in under-standing the distribution of this species [66]. Similarly, Catala-Roman et al. [67] used un-manned aerial system (UASYS) equipped with neural networks (CNNs) based on YOLOv10 for detecting and geo-locating two invasive species Araujia sericifera Brot. and Cortaderia selloana in the citrus orchard. The system significantly enhanced productivity, reduced the costs associated with manual weeding in organic farming and promotes en-vironmental sustainability, offering a scalable method for both organic and conventional agriculture.
Additionally, we have added some examples of different smart systems in the market in the revised manuscript (lines 194-210) to enhance the readability of the content of our manuscript.
Comments 12: Line 215 to 217, Identification accuracy is of course important. Yet, can the invasive species be effectively removed? For example, during the process of removal, escape is probable. Also, when removing invasive plant species, remaining tuber roots means that rapid return of pest species will happen.
Response 12: Thank you for the insightful question. While high identification accuracy is fundamental to successful invasive species management, effective removal is indeed the critical next step. Drones and automated systems are increasingly being integrated not just for detection, but also for precision-targeted removal strategies, although limitations such as escape and regrowth persist and must be addressed through multi-stage management approaches.
For instance, the risk of escape during removal is minimized when drones are coupled with automated herbicide applicators or biocontrol delivery systems that act immediately after detection. Real-time decision-making algorithms reduce delays and ensure rapid intervention. Research in Australia has demonstrated successful applications of drones equipped with infrared-guided systems for targeting invasive species such as Rhinella marina (cane toads), minimizing escape and improving control rates [65], (lines 174-177). In plant systems, precision herbicide delivery mechanisms, guided by real-time data from multispectral or hyperspectral imaging, ensure that only the invasive species is affected, preserving native vegetation while reducing the window for escape or re-establishment (lines 230-239).
Regarding regrowth from remaining tubers, rhizomes, or roots, which is a common issue in managing species like Reynoutria japonica or Cyperus rotundus, integrated approaches are essential. Single-pass removal is often insufficient, and the most effective strategies involve sequential monitoring and follow-up treatments. Smart systems can aid this by scheduling revisit operations based on machine learning-driven regrowth models, which predict the probability and location of re-emergence. Additionally, soil-penetrating radar (SPR) and thermal imaging sensors are being explored to detect below-ground biomass, increasing the efficacy of root-level removal.
Moreover, machine vision systems using deep learning, such as the work by Ferreira et al. [85], lines 252-257, not only achieve near-perfect classification accuracy (99.5%) but can also segment overlapping canopies and distinguish species in dense growth stages. This segmentation can be extended to map potential underground structures based on above-ground cues, aiding in comprehensive removal efforts.
Challenges and prospects of smart chip-enabled invasive plant management
Comments 13: Line 185 to 189, Other than illustrative arts, I believe that photographs of invasive species removal systems can be used. With proper citation, photographs can be inserted.
Response 13: Thank you for the suggestion. We have added the photographs of invasive species removal systems (Figure 3) with a citation and revised the section “Challenges and prospects of smart chip-enabled invasive plant management” in the manuscript (lines 292-298) by adding a little information about the removal systems of invasive plants.
Conclusion
Comments 14: I have no critical comments on this section of the manuscript.
Response 14: Thank you for your positive feedback.

Reviewer 3 Report
Comments and Suggestions for Authors
The manuscript 'Smart chip technology for the control and management of invasive plant species: a review' is a well prepared analysis.
This work investigates the use of smart chip technology (SCT) as a sustainable and accurate alternative to traditional invasive plant control methods, which are often inefficient and environmentally harmful. By integrating microchip sensors, AI, IoT and remote sensing, SCT enables real-time monitoring and predictive modeling. This approach enables targeted, environmentally friendly management strategies that reduce herbicide use and minimize ecosystem damage. While challenges remain, such as cost, biodegradability, and regulatory constraints, advances in biodegradable electronics and AI-based automation offer promising solutions.
My comments are below.
Be consistent when writing species names: use the scientific name followed by the common name.
For example, reference 16, line 58 does not exactly reflect the authors' statement.
line 101 novel or targeted?
References 55 and 56 are missing. Check them all.
Author Response
Response to Reviewer 3
This work investigates the use of smart chip technology (SCT) as a sustainable and accurate alternative to traditional invasive plant control methods, which are often inefficient and environmentally harmful. By integrating microchip sensors, AI, IoT and remote sensing, SCT enables real-time monitoring and predictive modeling. This approach enables targeted, environmentally friendly management strategies that reduce herbicide use and minimize ecosystem damage. While challenges remain, such as cost, biodegradability, and regulatory constraints, advances in biodegradable electronics and AI-based automation offer promising solutions.
Response: Thank you for your valuable feedback. Your suggestion helped us to improve the quality of our manuscript. All the changes have been made in manuscript highlighted in RED
My comments are below.
Be consistent when writing species names: use the scientific name followed by the common name.
Response: Thank for the suggestion. We thoroughly checked the manuscript, use the scientific name of the species followed by the common name.
For example, reference 16, line 58 does not exactly reflect the authors' statement.
Response: Thank you for your insightful observation. we have revised the statement such as “Smart chip technology (SCT) has already demonstrated its potential in precision agriculture, forestry, and conservation, particularly through its integration in sensor-based systems, data analytics, and automated monitoring [16] for greater accuracy and updated the manuscript (lines 65-66)
line 101 novel or targeted?
Response: We have updated line 109 as Integrating SCT into invasive plant management represents a novel approach to crop management.
References 55 and 56 are missing. Check them all.
Response: Thank you for highlighting. The information of ALL references has been added accurately and we checked them throughout the manuscript.
Round 2
Reviewer 3 Report
Comments and Suggestions for Authors
I noticed that the authors made some changes improving the quality of the manuscript. Unfortunately, the changes were not enough. The authors were superficial in this revision. There are comments from the first round to which they responded favorably but no changes are made throughout the manuscript.
I do not understand the point of figure 3. It is presented in detail and pictures one of the articles considered in this review. Why?
lines 293-299 revise the description of this article that has just been inserted.
Author Response
Response 1: First of all, we sincerely thank you for taking the time to review our manuscript. We apologize for any confusion or inconvenience caused during the first round of revision.
In response to your comments, we have carefully reviewed the manuscript and made substantial revisions while keeping the core contents of the manuscript. Additional improvements have been made (highlighted in Red in the revised manuscript) where the structure of the manuscript looks weak. Specifically, revisions have been made in the following sections:
Abstract: lines 24-27, lines 29-30
Introduction: lines 54-57, line 66-74, line 80-83.
Smart Chip Technology: An Innovative Approach: lines 103-116, lines 122-134, lines 154-155.
Smart Chip-Enabled Invasive Plant Management: lines 168-169, 175-180, lines 191-207, lines 260-262, and lines 280-281.
Challenges and prospects of smart chip-enabled invasive plant management: lines 288-293, lines 319-322, lines 324-327, 337-339.
In addition, typing errors throughout the manuscript, including those in the reference list, have been corrected and are also highlighted in red.
We hope these revisions adequately address your concerns and improve the overall quality of the manuscript.
The other changes have been during first round of revision in response to the reviewers' comments and suggestions, specifically:
- We have added Table S1, which lists a variety of commercial and research-based sensor models and chips commonly utilized in weed control and precision agriculture. This table includes details such as application area, functionality, data collection method, and relevant sources. The goal of this addition is to assist researchers and practitioners in selecting appropriate technologies for invasive species monitoring and other smart agricultural applications.
- We have updated the content in the revised manuscript (Lines 40–45) to include definitions from additional authoritative organizations, ensuring greater clarity and consistency.
- To improve reader understanding, we have further elaborated on the operational mechanism of bioengineered smart chips. In the revised version (Lines 122–134), we clarify that these chips typically function as part of an integrated biosensing system.
- In addition, we have added some examples of different smart systems available in the market in the revised manuscript (lines 191-207) to enhance the readability of the content of our manuscript and potentially help authors/researchers looking for information about the usefulness of the systems.
- We have also addressed other suggestions, including those related to species names and references, and have made the necessary corrections throughout the manuscript.
- Regarding Figure 3, this was included based on a reviewer's recommendation. The figure contains photographs of invasive species removal systems, with appropriate sources cited. In Lines 288–293, we have provided background information and linked the figure to the section titled “Challenges and Prospects of Smart Chip-Enabled Invasive Plant Management.”
We have taken great care to incorporate all reviewer feedback while maintaining the core content and structure of the manuscript. Additionally, any language issues have been addressed by one of our co-authors, who is a native English speaker.
Thank you again for your valuable insights and recommendations.

Round 3
Reviewer 3 Report
Comments and Suggestions for Authors
The modifications done by the authors have improved the quality of the ms. Therefore, I have no further comments and consider that it complies with the required standards for publication.